# Dopamine Inhibits Arabidopsis Growth through Increased Oxidative Stress and Auxin Activity

Timothy E. Shull, Jasmina Kurepa and Jan A. Smalle *

Department of Plant and Soil Sciences, University of Kentucky, Lexington, KY 40546, USA
* Correspondence: jsmalle@uky.edu

**Abstract:** Like some bacterial species and all animals, plants synthesize dopamine and react to its exogenous applications. Despite dopamine's widespread presence and activity in plants, its role in plant physiology is still poorly understood. Using targeted experimentation informed by the transcriptomic response to dopamine exposure, we identify three major effects of dopamine. First, we show that dopamine causes hypersensitivity to auxin indole-3-acetic acid by enhancing auxin activity. Second, we show that dopamine increases oxidative stress, which can be mitigated with glutathione. Third, we find that dopamine downregulates iron uptake mechanisms, leading to a decreased iron content—a response possibly aimed at reducing DA-induced oxidative stress. Finally, we show that dopamine-induced auxin sensitivity is downstream of glutathione biosynthesis, indicating that the auxin response is likely a consequence of DA-induced oxidative stress. Collectively, our results show that exogenous dopamine increases oxidative stress, which inhibits growth both directly and indirectly by promoting glutathione-biosynthesis-dependent auxin hypersensitivity.

**Keywords:** auxin; catecholamines; dopamine; glutathione; iron; reactive oxygen species





## 1. Introduction

Because dopamine (DA) plays an essential role in motor control, executive function, motivation, and reinforcement and reward in humans, DA is almost reflexively associated with humans or animals. However, DA biosynthetic pathways exist in plants [1]. Like DA synthesis in vertebrates, DA is synthesized from tyrosine either through hydroxylation (yielding L-DOPA) followed by decarboxylation or through decarboxylation followed by hydroxylation [1]. Endogenous DA has been detected in several plant species, and yet a consensus about the function of DA in plants has not been reached [1].

Before the discovery that the most important reward pathway in the mammalian brain is the mesolimbic dopamine pathway [2,3], it was hypothesized that DA protects plants from foraging animals [4]. Another potential explanation for the conservation of DA biosynthesis in plants is the apparent involvement of DA-phenylpropanoid conjugates in the pathogen response [5]. Others proposed that—like its precursor L-DOPA—DA functions as an allelochemical [6,7]. Contrasting the allelopathic hypothesis, exogenous DA promotes growth under stressful conditions [1,8]. More frequently reported is that DA affects plant growth by altering the activity of phytohormones [1,9–12]. Most notably, DA has been reported to inhibit the oxidation of auxin indole-3-acetic acid (IAA) both in vitro and in vivo by inhibiting IAA oxidases [9], which, in Arabidopsis, are encoded by members of the 2-oxoglutarate- and Fe(II)-dependent oxygenase superfamily genes *DIOXYGENASES FOR AUXIN OXIDATION (DAO)* [13].

Research on the dysfunction of dopaminergic neurons shows that both DA and DA metabolism affect cellular redox homeostasis [14]. DA (3-hydroxytyramine) is a catecholamine, and because its catechol (1,2-dihydroxy benzene) moiety can undergo a series of redox reactions, it is an anti- or a pro-oxidant depending on its concentration and molecular environment [14–16]. Knowing that DA can undergo spontaneous oxidation, which

is accelerated by redox-active metals (e.g., iron) and inhibited by antioxidants (e.g., glutathione (GSH)), DA is mostly viewed as a pro-oxidant [14,17–19]. The spontaneous metal- or enzyme-catalyzed oxidation of DA leads to the formation of damaging reactive oxygen species (ROS) and DA-quinones [14].

Here, we use the transcriptomic response to short-term DA treatments to generate hypotheses concerning the effect of DA on Arabidopsis plants. Using targeted experimentation based on our transcriptomics analysis, we show that DA acts as a pro-oxidant when exogenously applied. In addition to the growth-inhibitory effect of increased oxidative stress, we find that DA enhances auxin activity and dysregulates iron homeostasis.

## 2. Results

### 2.1. DA Inhibits Arabidopsis Growth and Causes Broad Transcriptomic Changes

To establish how DA influences Arabidopsis growth, we grew Col-0 seedlings on MS/2 media containing a range of DA concentrations (0 to 1 mM), and, after 14 days of growth, we collected rosette diameter and primary root length measurements (Figure 1a–d). The mean rosette diameter significantly decreased in the plants grown on media supplemented with 0.2 mM DA or higher, with rosettes measuring at ~39% of the control at a dose of 0.3 mM DA and stabilizing at ~23% of the control at doses above 0.5 mM (Figure 1a,b). The growth of the primary root was more sensitive to DA than rosette growth was, and a dose as low as 0.1 mM DA led to a significant reduction in root length (Figure 1c,d). Moreover, the number of adventitious roots significantly increased in the 0.1 mM DA-treated plants (Figure S1a), whereas lateral root growth was stimulated at even lower doses (e.g., 10 μM; Figure S1b). We also observed that—like in soybean [20]—DA or its derivatives are mobile within Arabidopsis seedlings, as evidenced by the accumulation of melanin in the shoots and roots of DA-treated plants (Figure 1e).

To start uncovering the molecular mechanisms that lead to DA-induced growth inhibition, we performed RNA-seq. To reduce the expression noise of developmentally regulated genes and to maximize the response, we used a high dose of 50 mM DA for short-term treatments of 2 and 8 h. We compared the transcriptomes of the DA-treated samples to the control at each timepoint and trimmed the genes with log2 fold changes (FC) between −1 and 1 and false discovery rates (FDRs) > 0.05, yielding 1152 differentially expressed genes (DEGs) (Table S1). To gain a broad overview of the impact of DA and to facilitate further exploration of our dataset, we conducted a Gene Ontology overrepresentation analysis (GOA) which revealed that DA induces a highly complex transcriptomic response and causes the upregulation of the genes associated with auxin and phytoalexin biosynthesis; stress-responsive hormones, such as abscisic, salicylic, and jasmonic acids; oxidative stress; and the plant immune response, among many others (Table S2). After 2 and 8 h of DA treatment, the ontology groups associated with growth and photosynthesis were overrepresented amongst the downregulated genes (Table S2).

To assess the most significantly impacted processes associated with DA-regulated DEGs, we examined the top four upregulated and downregulated genes at 8 h (Figure 1f). The most upregulated DEG was *ABCG40*, encoding an ABC G group transporter known to be upregulated in response to various stresses [21–24]. The second highly upregulated gene, *bHLH100*, encodes a basic helix-loop-helix transcription factor that regulates iron-deficiency responses and iron distribution [25]. Interestingly, *bHLH101*, the functional homolog of bHLH100 [25], was upregulated by the DA precursor L-DOPA [26]. The third strongly upregulated DEG was *GSTU11*, and it encodes a plant-specific tau class glutathione-S-transferase, an enzyme involved in the biosynthesis of defense-related glucosinolates [27]. Finally, we found that *CYP71A13*, which encodes a key cytochrome P450 enzyme in the phytoalexin/camalexin biosynthesis pathway [28,29], was strongly upregulated by DA. This gene was also shown to be upregulated by L-DOPA [26]. The gene encoding peroxidase isoform 45 (PER45) was downregulated by DA. *PER45* was also repressed by isothiocyanates (the degradation products of glucosinolates) [30] and auxin [31]. Like *PER45*, the strongly DA-downregulated *At4g15390* was also shown to be downregulated

by auxin [31]. The last two strongly DA-downregulated genes are involved in iron homeostasis. The first gene, *FER1*, encodes the iron-binding protein ferritin, which manages the cellular levels of reactive iron [32], and was also downregulated by L-DOPA [26]. The second gene, *ENH1* encodes an iron-binding rubredoxin family protein involved in abiotic stress responses [33]. Another rubredoxin family member was shown to be strongly down-regulated by L-DOPA [26]. Collectively, our targeted examination of highly differentially regulated genes in combination with our GOA indicated that DA alters the oxidative stress status, iron homeostasis, and auxin activity.

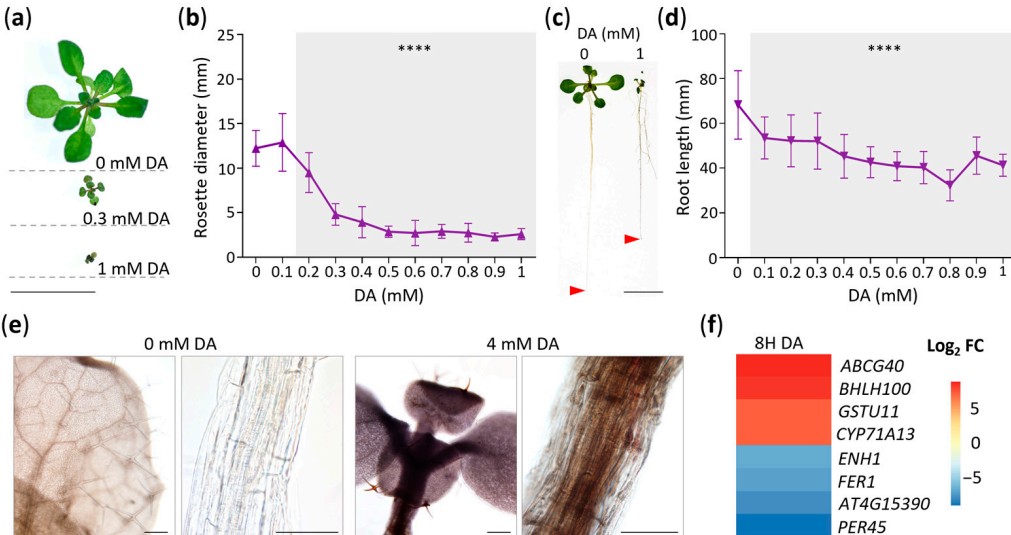

**Figure 1.** Dopamine (DA) alters plant growth and development. (**a**) Photographs of rosettes of representative 14-day-old seedlings grown on horizontally positioned petri plates with MS/2 media supplemented with the denoted concentrations of DA. Scale bar: 10 mm. (**b**) Rosette diameter of seedlings grown as in (**a**). Seedlings were photographed and rosette diameter was measured using ImageJ software. Data are presented as mean $\pm$ SD ($n \geq 28$; **** represents $p < 0.0001$; one-way ANOVA with Dunnett's post-test comparing DA treatments to control; shaded area demarks significant results). (**c**) Photographs of 14-day-old seedlings grown on vertically positioned petri plates with MS/2 media supplemented with the denoted concentrations of DA. Representative seedlings were moved to fresh plates for photography. Scale bar: 5 mm. (**d**) Primary root length of seedlings grown as in (**c**). Seedlings were photographed, and root length was measured using ImageJ software. Data are presented as mean $\pm$ SD ($n \geq 20$; **** represents $p < 0.0001$; one-way ANOVA with Dunnett's post-test comparing DA treatments to control; shaded area demarks significant results). (**e**) Arabidopsis seedlings grown in the presence of DA accumulate melanin in roots and shoots. Col-0 was grown vertically on MS/2 media supplemented with 0 or 4 mM DA for 14 days. Plants were cleared in 100% EtOH and were then rehydrated in 50% glycerol. Scale: 100 μm. Shoot and root micrographs of each treatment are depicted. (**f**) Top dopamine-responsive genes suggest disrupted iron homeostasis, altered IAA metabolism, and oxidative stress. Gene-level expression data were generated as described in the Materials and Methods, and the top 4 upregulated and downregulated genes at 8 h (8H DA) are presented as a heatmap of log2 fold change (FC). All presented DEGs with a non-zero log2 FC have an FDR < 0.001.

### 2.2. DA Does Not Increase IAA Accumulation

The strong upregulation of *CYP71A13* (Figure 1f) and our GOA (Table S2) indicated that DA impacts aspects of auxin metabolism. Thus, we then analyzed this metabolic pathway to build on the previous study that showed that DA induces IAA hypersensitivity by inhibiting IAA oxidases [9]. To that end, we examined the DEGs encoding the enzymes involved in IAA and phytoalexin metabolism [34,35] and measured the levels of different auxin species using targeted metabolomic analyses (Figure 2). Transcriptomic analyses showed that all the genes (*CYP79B3*, *CYP79B2*, *CYP71A13*, and *CY71A12*) encoding the

enzymes that catalyze the synthesis of indole-3-acetonitrile were upregulated by the DA treatment, as were the genes encoding the phytoalexin synthesis enzymes (*CYP71B6*, *AO1*, *GGP1*, *PAD3*, and *PAD4*) (Figure 2). In addition, DA promoted the upregulation of the genes encoding the enzymes capable of producing IAA (namely, *NIT4* and *NIT2*, which are thought to form IAA directly from indole-3-acetonitrile or through the intermediate indole-3-acetimide [36]) and the genes encoding the enzymes that form IAA conjugates (*GH3.2* and *GH3.3*), which indicated that both phytoalexins and IAA might accumulate in response to DA. Importantly, the expression of the *DAO* genes was not affected by DA. Also noteworthy is the lack of a significant impact of DA on the expression of *TAA1* and the *YUCCA* gene family, which encode the enzymes of the indole-3-pyruvate branch of IAA biosynthesis, from which the majority of IAA is produced in most plants [36].

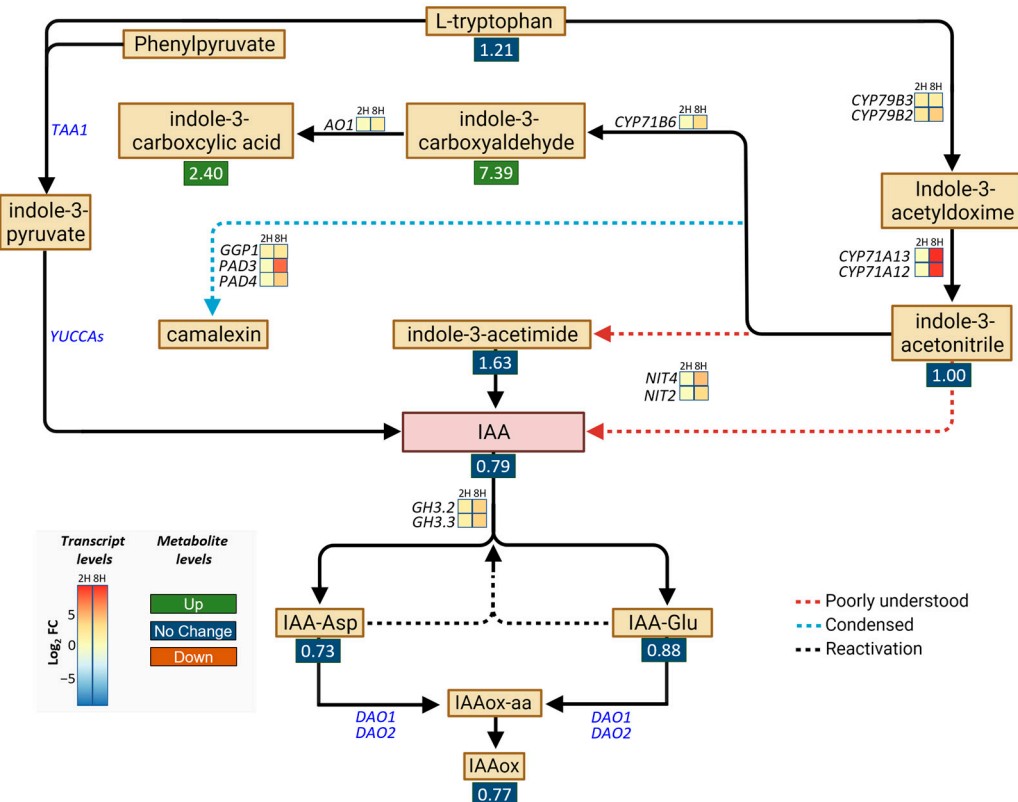

**Figure 2.** Map of differentially expressed genes (DEGs) and metabolites involved in IAA metabolism. DEGs and their relative expression levels (presented as log2 fold changes (FC)) in seedlings treated for 2 h (2H) or 8 h (8H) with 0 or 50 mM DA are shown as heatmaps. All presented DEGs with a non-zero log2 FC have an FDR < 0.001. The FC in metabolite level in seedlings grown on 0.5 mM DA for 7 days is presented below the metabolites in colored boxes (green, FC ≥ 2, accumulation; red, FC ≤ 0.5, reduced levels; blue, 0.5 ≥ FC ≤ 2, not altered). Red arrows represent poorly understood processes, light-blue dashed arrows represent condensed metabolic steps, and black dashed arrows represent the reactivation of IAA through deconjugation of IAA amino conjugates. Abbreviations: IAA represents indole-3-acetic acid, IAA-Asp represents aspartate-conjugated IAA, IAA-Glu represents glutamine-conjugated IAA, IAAox-aa represents oxidized amino-conjugated IAA, and IAAox represents oxidized IAA.

Considering the identity and the number of auxin metabolic genes that were DA-responsive in the short-term treatments, we next opted to determine if auxin species differentially accumulate in plants grown on DA-treated media. To that end, extracts of Col-0 seedlings grown on control or 0.5 mM DA-supplemented media for 7 days were analyzed using UPLC-ESI-MS/MS. Metabolomic analyses confirmed that the DA treatments led to the accumulation of phytoalexins (the indole-3-carboxaldehyde FC = ~7.39, and the

indole-3-carboxycylic acid FC = ~2.40; Figure 2, Table S3). Since both biotic and abiotic stresses can trigger increased phytoalexin biosynthesis [37], we concluded that DA exposure causes a broad stress response in Arabidopsis. We observed no noteworthy changes in the IAA, IAA-Glu, IAA-Asp, or IAAox levels (Figure 2, Table S3), which was contrary to the expectation that DA would increase IAA accumulation by inhibiting IAA oxidases [38]. In addition, as tryptamine was undetectable in the DA-treated samples, its levels may be negatively regulated by DA treatment (Table S3). Drawing from our transcriptomic and metabolomic data, as well as the current models of auxin metabolism in Arabidopsis [13], we concluded that DA treatment does not increase IAA levels either by increasing IAA synthesis or by decreasing the rate of IAA oxidation.

*2.3. DA Increases Auxin Action*

Since some aspects of the response of plants to DA were reminiscent of auxin-related phenotypes (e.g., decreased root length and altered lateral root formation [39,40]) and since one of the highly downregulated genes presented in Figure 1f (*AT4G15390*) was also downregulated by auxin, we continued analyzing DA's impact on auxin activity. To that end, we employed the auxin-inducible *DR5::GUS* reporter line and found that seedlings grown on DA-supplemented media express more *GUS* in both primary and lateral root tips compared to the control (Figure 3a). This was independently confirmed using a transgenic line that contained the auxin-inducible reporter *DR5::RFP*. *RFP* expression in the root tips of the plants grown on the DA-supplemented media was ~2.4 times higher than that of the control (Figure 3b,c). Taken together, we concluded that DA increases auxin activity. Supporting this conclusion, out of the 24 auxin-regulated DEGs identified in our transcriptomics analysis, 87% were upregulated at 2 h, and 71% were upregulated at 8 h (Figure S2a).

We next hypothesized that DA-induced auxin activity is responsible for the growth-inhibitory action of DA. To test this, we grew the auxin-resistant mutants *axr2-1*, *axr3-3*, and *axr5-1* on DA-supplemented media for 14 days and found that all three mutants had a mild but significant tolerance to the inhibitory effects of DA on rosette growth at 0.4 and 0.5 mM doses (Figure S2b). To further explore the DA-enhancing effect on auxin activity, we used root elongation as a metric for auxin sensitivity. We tested the effects of DA + IAA co-treatments by measuring the root length of the Col-0 seedlings grown on the media with 0, 10 nM, or 100 nM IAA combined with no DA or 0.1 mM DA (Figure S2c). After 11 days of growth, none of the low-concentration IAA treatments had an impact on root length, whereas the 0.1 mM DA treatment led to an expected small reduction in root length (Figure S2c). However, the IAA and DA co-treatments had a potent synergism: the plants grown on the media supplemented with 0.1 mM DA + 10 nM IAA had mean root lengths of ~33%, and the plants grown on 0.1 mM DA + 100 nM IAA had mean root lengths of ~12% compared to the DA treatment alone (Figure S2c).

To evaluate if this synergistic effect was due to increased auxin signaling, we assessed its impact on the Col-0 wild type and the severe auxin-resistant mutant *axr3-3* [41] (Figure 3d–f). After 11 days of growth, the rosettes and roots of the wild-type seedlings grown on the 0.1 mM DA + 100 nM IAA-supplemented media were significantly growth-inhibited compared to the DA-only treatment (rosette growth was ~73% of the control, and root growth was ~14% of the control), whereas the growth of the axr3-3 roots and shoots was not inhibited but was significantly increased instead (rosette growth was ~138% of the control, and root growth was ~137% of the control; Figure 3d–f). We concluded that DA increases the sensitivity of Arabidopsis to exogenous IAA and that this increased sensitivity involves the canonical auxin response pathway.

*2.4. DA-Enhanced Auxin Activity Requires the Auxin Transporter AUX1*

DA altered the expression of genes encoding components of auxin transport machinery (Figure S3, Table S2). To further explore this, we incorporated the use of the synthetic auxin 2-naphthaleneacetic acid (NAA) in our combinational treatments. Because IAA enters

roots solely through active uptake and NAA passively diffuses into roots, these two auxins were used to identify responses that depend on auxin transport [42]. Contrary to the combinatorial effect of DA + IAA co-treatments, the response of DA + NAA co-treated plants were comparable to the response of the plants treated with NAA only (Figure 4). For example, the mean root length of the seedlings treated with 10 μM DA + 10 nM IAA for 14 days was 37% of the mean root length of the seedlings treated with 10 μM DA only, whereas the mean root length of the 10 μM DA + 10 nM NAA treated plants was not significantly different from that of the NAA-only control (Figure 4a,b). The 10 μM DA + 10 nM NAA co-treated plants had a wild-type response even at the highest tested DA concentration (100 μM; Figure 4b). Similarly, whereas the 10 μM DA + 100 μM IAA treatments led to a significant increase in the mean number of adventitious roots (Figure 4c) and mean root hair length (Figure 4d,e), when compared to the IAA-only control, the DA + NAA treatments did not lead to a significant change in any of these parameters compared to the NAA-only control. Therefore, although 10 nM NAA caused a stronger growth response than 10 nM IAA (e.g., significantly shorter roots (Figure 4a,b), caused more adventitious roots (Figure 4c), and longer root hairs ($p < 0.0001$, Figure 4e), we observed no DA-dependent hypersensitization in the NAA-treated seedlings, which strongly suggests that the DA/IAA interaction requires auxin transport machinery.

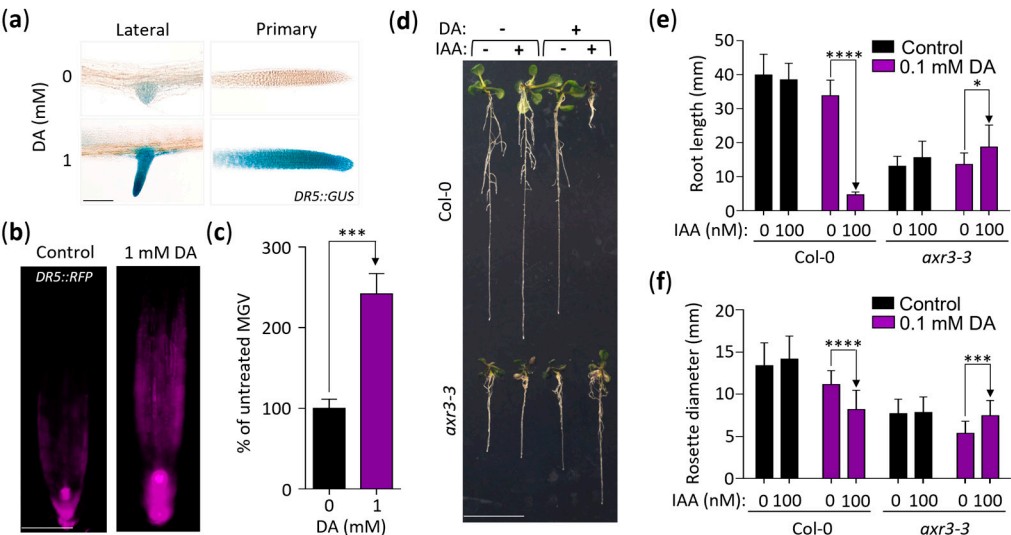

**Figure 3.** DA enhances auxin activity in roots and shoots. (**a**) *DR5::GUS* expression is higher in the root tips of seedlings grown on DA-supplemented media. Seedlings grown for 7 days on control and 1 mM DA-supplemented MS/2 media were immersed in a buffer containing 1 mg/mL of X-Gluc, were cleared in 100% ethanol, and were rehydrated in 50% glycerol. Micrographs were taken at 20× magnification. Scale: 100 μm. (**b**) *DR5::RFP* expression confirms that DA alters auxin activity in roots. *DR5::RFP* seedlings were grown as in (**a**), and RFP signal was captured using fluorescence microscopy. Scale: 100 μm. (**c**) Quantification of DA-induced *DR5::RFP* expression in root tips shown in (**b**). Mean grey value (MGV) of root tips was measured up to 600 μm from the root cap. Data are presented as percent change in MGV ± SEM ($n \geq 6$). *** represents $p < 0.001$ (one-way ANOVA with Dunnett's post-test; comparisons are labeled with one-way arrows). (**d**) Auxin-insensitive plants resist DA-induced IAA hypersensitivity. Col-0 and *axr3-3* seedlings were grown vertically on media with or without 100 μM DA and 100 nM IAA for 11 days. Representative seedlings were transferred to a fresh plate and were photographed. Scale bar: 10 mm. (**e,f**) The impact of DA-induced IAA hypersensitivity on primary root growth and rosette size. Plants were grown as in (**d**) and were photographed, and root length and rosette diameter were measured using ImageJ software. Data are presented as mean ± SD ($n \geq 16$; * represents $p < 0.05$; *** represents $p < 0.001$; **** represents $p < 0.0001$; one-way ANOVA with Tukey's post-test; multiple comparisons are denoted by one-way arrows).

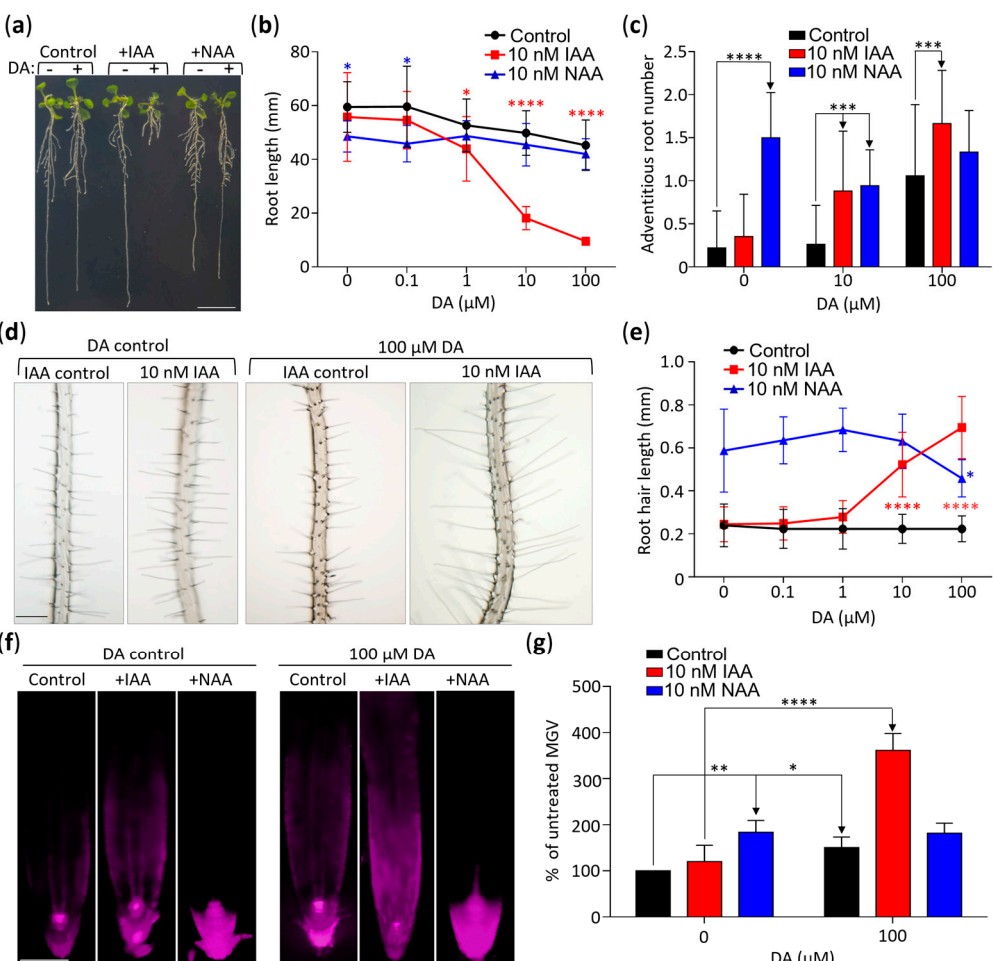

**Figure 4.** DA-dependent hypersensitization of IAA responses is linked to auxin influx. (**a**) DA exposure causes hypersensitivity to IAA but not to NAA. Representative Col-0 seedlings grown vertically for 14 days on media, were transferred to a fresh plate, and were photographed. Scale: 10 mm. (**b**) Quantitative analysis of the combinational impact of DA, IAA, and NAA on primary root length. Plants were grown as described in (**a**) and were photographed, and primary root length was measured using ImageJ software. Data are presented as mean ± SD ($n \geq 12$; * represents $p < 0.05$; **** represents $p < 0.0001$; two-way ANOVA with Tukey's post-test comparing the control with hormone treatments; significance markers are color-coded by treatment to highlight comparisons to control). (**c**) Quantitative analysis of the combinational impact of DA, IAA, and NAA on the number of adventitious roots. Adventitious roots of plants grown as in (**a**) were counted using a microscope. Data are presented as mean ± SD ($n \geq 16$; *** represents $p < 0.001$; **** represents $p < 0.0001$; two-way ANOVA with Tukey's post-test; multiple comparisons are denoted by one-way arrows). (**d**) DA and IAA co-treatment increases root hair length. Micrographs of plants grown as in (**a**) were taken at 4× magnification. Scale: 100 μm. (**e**) Graph illustrating DA's effect on IAA-induced increase in root hair length. Root hairs were imaged as in (**d**), and their length was measured using ImageJ software. Data are presented as mean ± SD ($n \geq 100$; * represents $p < 0.05$; **** represents $p < 0.0001$; Tukey's post-test comparing control with hormone treatments; significance markers are color-coded by treatment to highlight comparisons to control). (**f**) *DR5::RFP* expression confirms that DA induces hypersensitivity to IAA but not to NAA. *DR5::RFP* seedlings were grown for 7 days at the denoted concentrations of DA, IAA, and NAA, and RFP fluorescence was captured using fluorescent microscopy. Scale: 100 μm. (**g**) Quantitative analysis of *DR5::RFP* signal shown in (**f**). Root tissue was traced 600 μm basipetally from the root cap, and mean grey value (MGV) was calculated using ImageJ software. Data are presented as percent of the control MGV ± SEM ($n \geq 12$; **** represents $p < 0.0001$; ** represents $p < 0.01$; * represents $p < 0.05$; two-way ANOVA with Tukey's post-test; multiple comparisons are marked with one-way arrows).

Additional evidence that DA does not hypersensitize plants to NAA was obtained through an analyses of *DR5::RFP* seedlings grown for 7 days on media supplemented with 10 nM IAA, 100 μM DA + 10 nM IAA, 10 nM NAA, and 100 μM DA + 10 nM NAA (Figure 4f,g). As expected from the root growth response assays, the DA, NAA, and DA + IAA treatments led to significant increases in RFP fluorescence compared to their respective controls, whereas the DA + NAA co-treated plants did not differ from the NAA-only control (Figure 4f,g).

Lastly, we analyzed the DA-induced IAA hypersensitivity response in the *aux1-7* mutant (Figure 5), which is deficient in the activity of the major auxin influx carrier AUX1, whose corresponding gene was downregulated in the DA-treated plants (Figure S3) [43–46]. When wild-type (Col-0) and *aux1-7* seedlings were grown for 11 days on vertically positioned plates with control or 10 μM DA + 10 nM IAA-supplemented media, we observed the expected DA-hypersensitization of the auxin root growth response in the wild type but no such response in the *aux1-7* seedlings (Figure 5a–c). Next, we tested the expression of *DR5::GUS* in plants grown on 10 μM DA + 10 nM IAA-supplemented media and recorded the DA-induced hypersensitivity response in the wild-type plants but, again, not in the *aux1-7* plants, which exhibited more diffuse GUS expression in the mid-root instead (Figure 5d,e). We concluded that the DA-hypersensitization of the auxin response requires the AUX1-dependent auxin transport mechanism.

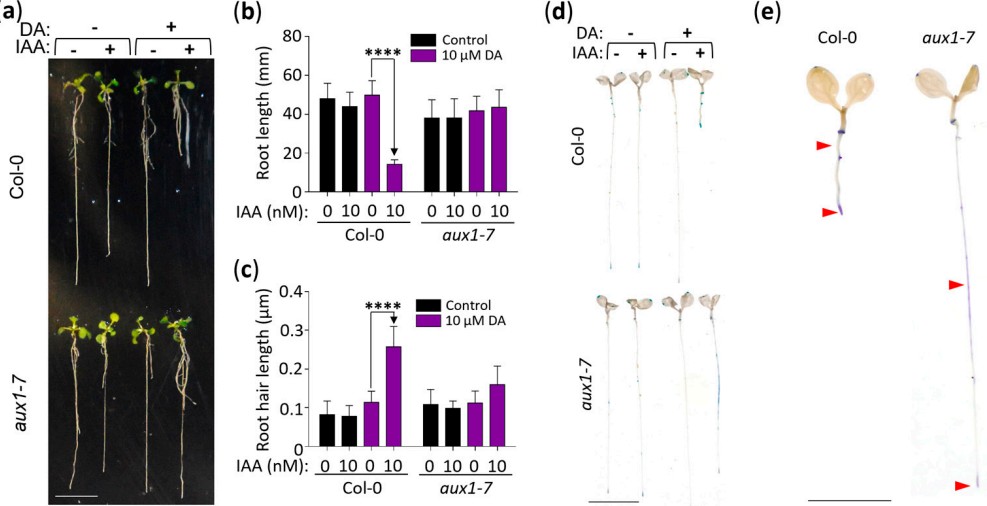

**Figure 5.** DA-induced IAA hypersensitivity depends on the auxin transporter AUX1. (**a**) Representative seedlings illustrating the differential impact of DA/IAA co-treatments on Col-0 and *aux1-7* root growth. Col-0 and *aux1-7* grown vertically for 11 days on the denoted media were transferred from the test plates to a fresh plate for photography. Scale: 10 mm. (**b**) Quantitative analysis of the differential impact of DA/IAA co-treatments on Col-0 and *aux1-7* root length. Plants were grown as in (**a**) and were photographed, and root length was measured using ImageJ software. Data are presented as mean $\pm$ SD ($n \geq 12$; **** represents $p < 0.0001$; two-way ANOVA with Tukey's post-test; multiple comparisons are demarked by one-way arrows). (**c**) Quantitative analysis of the differential impact of DA/IAA co-treatments on Col-0 and *aux1-7* root hair length. Plants were grown as described in (**a**), and micrographs of root hairs were taken at 4× magnification. Root hair length was measured using ImageJ software. Data are presented as mean $\pm$ SD ($n \geq 80$; **** represents $p < 0.0001$; two-way ANOVA with Tukey's post-test; multiple comparisons are demarked by one-way arrows). (**d**) *DR5::GUS* expression in DA/IAA co-treated plants is repressed in the *aux1-7* background. Seedlings grown for 7 days on denoted supplemented media were removed from the media, stained, dehydrated, rehydrated in 50% glycerol, and photographed. Scale: 5 mm. (**e**) Digitally excised images of Col-0 and *aux1-7* grown on media supplemented with 10 μM DA and 10 nM IAA. Red arrows point to areas of differential GUS staining. Scale: 5 mm.

## 2.5. DA-Induced Oxidative Stress Elevates the Antioxidant Glutathione

Our GOA revealed that DA treatments change the expression of several genes associated with the response to oxidative stress (Table S2). Considering that GSH is a central antioxidant, one of the most DA-responsive genes in our RNA-seq dataset was *GSTU11* (log2 FC of ~8.34), and previous studies revealed a link between GSH and auxin regulation [47,48], we analyzed the effects of DA on GSH metabolism. First, we examined the DA-responsive DEGs involved in GSH metabolic processes (Figure 6a). We found that, along with *GSTU11*, 11 other GSH transferases and *GSH2*, which encodes a GSH synthetase enzyme [49,50], were upregulated in response to DA treatment (Figure 6a), suggesting that GSH plays a role in the response of Arabidopsis to DA.

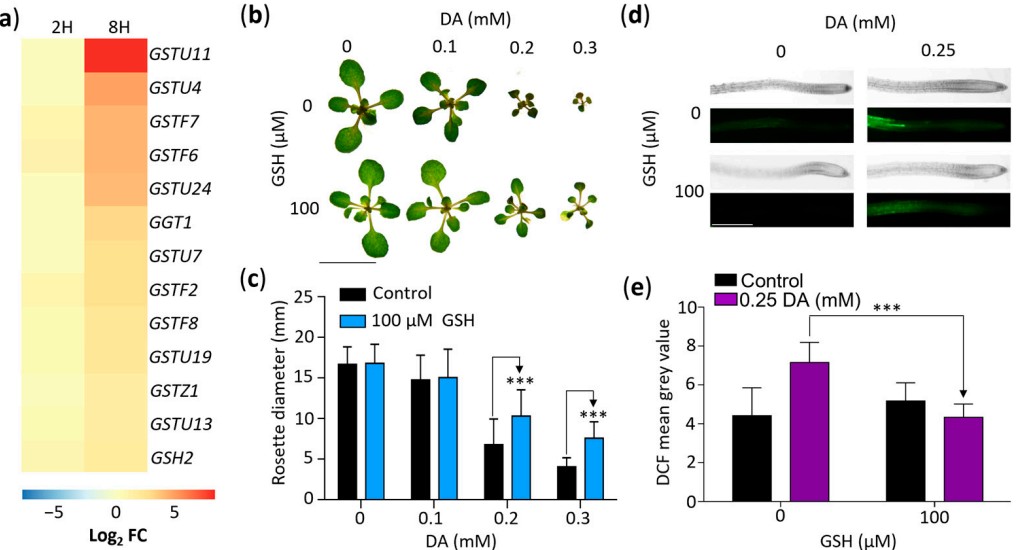

**Figure 6.** Glutathione (GSH) suppresses DA-induced growth inhibition and ROS accumulation. (**a**) DA upregulates the expression of genes encoding GSH transferases. Gene-level expression data were generated as described in the Materials and Methods. GSH-associated DEGs were extracted and are presented as a heatmap of log2 fold changes (Log2 FC) at 2 (2H) and 8 (8H) hours. All presented DEGs with a non-zero log2 FC have an FDR < 0.0001. (**b**) GSH supplementation attenuates DA-induced inhibition of rosette growth. Col-0 was grown for 14 days on MS/2 media supplemented with the denoted concentrations of DA and GSH. (**c**) Quantitative analysis of the impact of GSH supplementation on DA-induced rosette growth inhibition. Plants were grown as described in (**b**) and were photographed, and rosette diameter was measured using ImageJ software. Data are presented as mean ± SD (*n* ≥ 20; *** represents *p* < 0.001; one-way ANOVA with Tukey's post-test; multiple comparisons are demarked by one-way arrows). (**d**) GSH attenuates DA-induced ROS accumulation in roots. Col-0 seedlings grown for 7 days on MS/2 media supplemented with the denoted concentrations of DA and GSH were submerged in 10 μM H2DCF-DA and were incubated at room temperature in complete darkness for 15 min. DCF was visualized using fluorescent microscopy. Scale: 0.5 mm. (**e**) Quantitative analysis of the micrographs described in (**d**). Mean grey value (MGV) of DCF fluorescence was measured 1.5 mm from root cap. Data are presented at mean ± SD (*n* ≥ 9; *** represents *p* < 0.001; one-way ANOVA with Tukey's post-test; comparisons are demarked by one-way arrows).

To determine if the effect of short-term DA treatments on the regulation of genes associated with GSH metabolism is representative of a protective response to DA-generated oxidative stress, we grew plants for 14 days on media supplemented with either 0 or 100 μM GSH and 0, 0.1 mM, 0.2 mM, or 0.3 mM DA (Figure 6b,c). Indeed, the mean rosette diameter of the seedlings grown on the 0.2 mM DA + 100 μM GSH media was ~52% larger than that of the seedlings grown on the 0.2 mM DA-only media, and the mean rosette diameter of the 0.3 mM DA + 100 μM GSH seedlings showed an ~87% increase compared to the 0.3 mM DA-only control (Figure 6b,c). Therefore, such as in animal systems [51,52], GSH reduces the deleterious effect of DA treatments in plants, and plants respond to DA exposure by upregulating the genes associated with their GSH-dependent antioxidant defense system.

Lastly, to check if GSH protects plants from DA by mitigating ROS accumulation, we compared the ROS levels in control and DA-treated plants using 2′,7′-dichlorodihydrofluorescein diacetate (H2DCF-DA), a fluorogenic probe which reacts with ROS to produce highly fluorescent 2′,7′-dichlorofluorescein (DCF). Analyses of plants grown for 7 days on control or on 0.25 mM DA-containing media with or without 100 μM GSH showed stronger DCF fluorescence in the seedlings grown on the DA-containing media compared to the no-DA control, establishing that DA treatment indeed increases the endogenous ROS load in Arabidopsis cells. The DCF fluorescence levels in seedlings grown on DA + GSH were significantly lower compared to the DA-only control (Figure 6d,e). Therefore—like other organisms—[51,52] plants exposed to DA accumulate excess ROS, which leads to oxidative stress and a concomitant increased expression of the genes associated with GSH biosynthesis and metabolism. We concluded that, as documented in other plant species [53], DA stimulates GSH production in DA-treated seedlings.

## 2.6. DA-Enhanced Auxin Activity Depends on Glutathione Synthesis

The interactions between auxin and root redox status during root development is amply documented [47,54–57]. In addition, auxin activity is affected in GSH-deficient plants, thus establishing a link between auxin's control of root growth and GSH metabolism [47,57,58]. We examined the role of GSH biosynthesis in DA-induced changes in auxin activity using buthionine sulfoxamine (BSO), which decreases cellular GSH levels by inhibiting γ-ECS, the enzyme that catalyzes the first committed step of GSH biosynthesis [56,59]. We grew plants on media with 0, 0.5 mM, or 0.75 mM BSO with or without 10 μM DA and 10 nM IAA (Figure 7a,b). As reported previously [48], the mean primary root length of the seedlings grown on the 0.5 mM and 0.75 mM BSO media was significantly decreased compared to the control (~80% and ~65% of the control for 0.5 mM and 0.75 mM BSO, respectively). The addition of DA to the BSO media led to a complete recovery of primary root growth (Figure 7a,b), indicating that DA indeed stimulates GSH production in Arabidopsis. The average root lengths of the seedlings grown on the 0.5 mM and 0.75 mM BSO media with IAA were not significantly different from the BSO-only control. However, analyses of the plants grown on the media containing BSO, DA, and IAA showed that the inhibition of GSH biosynthesis partially but significantly reversed DA-induced IAA hypersensitivity. The mean root length of the seedlings grown on the 0.5 mM BSO-supplemented DA + IAA media showed an ~102% increase in primary root length compared to the DA + IAA co-treated plants (Figure 7a,b), indicating that DA-induced IAA hypersensitivity is—at least partially—a downstream effect of DA-induced GSH biosynthesis. The higher BSO dose (0.75 mM) was less effective at attenuating DA-induced IAA hypersensitivity (mean root length showed a 59% increase compared to the DA + IAA co-treated plants), which was not surprising considering that the effect of BSO on auxin activity is dose-dependent [48].

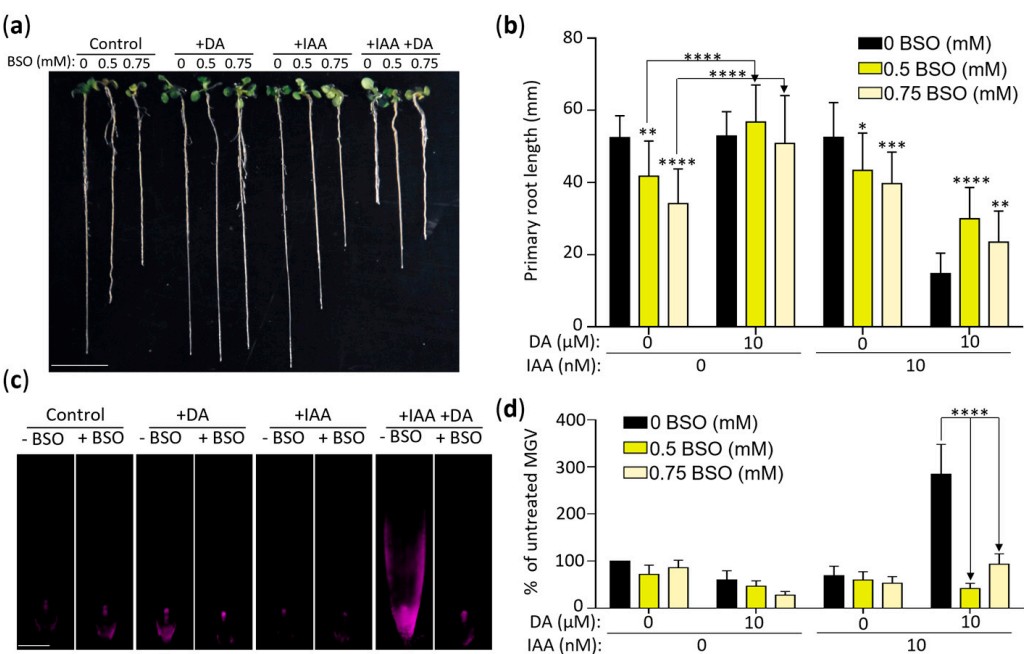

**Figure 7.** DA-induced auxin sensitivity is modulated by buthionine sulfoxamine (BSO). (**a**) Inhibition of GSH biosynthesis alters the impact of DA on root growth and auxin sensitivity. Col-0 seedlings grown vertically on MS/2 media supplemented with the denoted concentrations of BSO, DA (+DA, 10 μM), and IAA (+IAA, 10 nM) for 11 days were arranged and photographed. Scale: 10 mm. (**b**) Quantification of the inhibitory effect of BSO on DA-induced IAA hypersensitivity. Seedlings were grown as described in (**a**) and were photographed, and rosette diameter was measured using ImageJ software. Data are presented as mean ± SD ($n \geq 21$; **** represents $p < 0.0001$; *** represents $p < 0.001$; ** represents $p < 0.01$; * represents $p < 0.05$; two-way ANOVA with Tukey's post-test; stars without arrows are BSO treatments compared to the control; other comparisons are demarked by one-way arrows). (**c**) *DR5::RFP* expression confirms that BSO reduces DA-induced IAA hypersensitivity. *DR5::RFP* seedlings were grown for 7 days on the denoted concentrations of BSO, DA (+DA, 10 μM), and IAA (+IAA, 10 nM), and RFP fluorescence was captured using fluorescent microscopy. Scale: 100 μm. (**d**) Quantitative analysis of *DR5::RFP* signal described in (**c**). Root tissue was traced 600 μm from the root cap, and mean grey value (MGV) was calculated using ImageJ software. Data are presented as percent of the control MGV ± SEM ($n \geq 16$; **** represents $p < 0.0001$; two-way ANOVA with Tukey's post-test; multiple comparisons demarked by one-way arrows).

We confirmed the co-treatment growth response results at the auxin-induced gene expression level by using the *DR5::RFP* plants grown for 7 days on 0, 0.5 mM, or 0.75 mM BSO supplemented with DA and/or IAA (Figure 7c,d). DA and IAA co-treatments led to an expected significant increase in *DR5::RFP* expression in the BSO-free media ($p < 0.0001$ compared to the no-DA control). In contrast, adding BSO into the media led to a decrease in RFP fluorescence, signifying the restoration of normal auxin sensitivity levels. Therefore, both the DA-dependent reversal of BSO-induced root growth inhibition and the BSO-dependent attenuation of DA-induced IAA hypersensitivity indicate that DA promotes IAA hypersensitivity, in part, by increasing GSH biosynthesis.

### 2.7. DA-Induced Oxidative Stress Suppresses Iron Accumulation

Our results so far indicate that DA affects Arabidopsis growth by increasing oxidative stress, which leads to the upregulation of oxidative stress responses, including an increase in GSH biosynthesis that, in turn, enhances auxin activity. While the genes involved in iron ion homeostasis were not overrepresented in our GOA analysis (Table S2), identifying the iron homeostasis regulator encoding gene *BHLH100* as one of the genes most upregulated by DA (log2 FC of ~9.44; Figure 1f) suggested that a third potential growth-inhibitory effect of DA may be the dysregulation of iron homeostasis. To assess the extent of DA-induced changes in the regulation of iron homeostasis, we first selected the DA-induced DEGs associated with iron uptake and distribution (Figure 8a). Two DEGs were significantly downregulated by DA: the previously mentioned *FER1* (FC at 2H of −1.6, and FC at 8H of −4.3) and *FRO6* (FC at 2H of 0, and FC at 8H of −2.18), which encodes a ferric chelate reductase (FCR) involved in the maintenance of foliar iron homeostasis [60]. In addition to *BHLH100*, the upregulated gene set included *FRO3* (FC at 2H of 1.8, and FC at 8H of 2.9), encoding a mitochondrial *FCR* [61]; *NRAMP4* (FC at 2H of 1.8, and FC at 8H of 2.2), which regulates vacuolar iron stores [62]; *OPT3* (FC at 2H of 2.2, and FC at 8H of 2.3), encoding a phloem-specific iron transporter [63]; and *Brutus* (BTS; FC at 2H of 2.3, and FC at 8H of 3.5), which encodes an E3 ligase that negatively regulates iron uptake [64].

Considering this, we proceeded with targeted experimentation to determine if iron homeostasis is dysregulated in plants grown in the presence of DA. We measured the net effect of DA on iron homeostasis by measuring the iron content of the Col-0 seedlings grown for 7 days on the control media and on the 0.5 mM DA-supplemented media. The DA-grown seedlings accumulated ~33% less iron than those grown on the control media (Figure 8b). Consistent with that finding, immunoblotting analyses using the Col-0 seedlings grown on the 0, 0.25 mM, and 0.5 mM DA-supplemented media for 7 days showed that the level of FIT, a transcriptional regulator that plays a dominant role in sustaining the iron content [65], was reduced in the DA-grown plants (~0.45 and ~0.14 FC for 0.25 mM and 0.5 mM DA, respectively; Figure 8c), indicating that seedlings grown on DA-supplemented media accumulate less iron because of decreased uptake. Independent confirmation of this conclusion was obtained by measuring the activity of FCRs, enzymes that are required for the uptake of soluble iron [61]. FCR activity was significantly downregulated by DA treatment (Figure 8d).

Since iron deficiency severely affects plant growth [66], we hypothesized that if decreased iron uptake plays a key role in DA-induced growth inhibition, plants grown on media containing lower-than-optimal levels of iron should be hypersensitive to DA and that plants grown on media containing higher-than-optimal iron levels should be less affected by DA. To test this hypothesis, we grew plants on iron-free media supplemented with 5 (low), 50 μM (optimal), and 150 μM (high) Fe-EDTA with or without 0.1 mM DA for 14 days. As shown earlier (Figure 1b), the plants grown on the media containing optimal iron levels did not have decreased rosette diameters at 0.1 mM DA. The plants grown on the media containing low levels of iron were chlorotic with significantly reduced rosette diameters (~69% of the control, $p < 0.005$, Tukey's post-test, Figure 8e,f). However, the low- and high-iron experiments refuted our starting hypothesis: rather than being hypersensitive to DA, the seedlings grown on the low-iron media supplemented with DA were not distinguishable in size from those grown on the media with optimal iron levels, indicating that DA counteracts the negative effect of Fe deficiency on Arabidopsis growth. In contrast, while the seedlings grown on the high-iron media had significantly reduced rosette diameters compared to the plants grown on the media containing optimal levels of iron ($p < 0.001$, Tukey's post-test), the additional supplementation of the high-iron media with DA led to DA-hypersensitivity, suggesting that Fe enhances DA toxicity (Figure 8e,f).

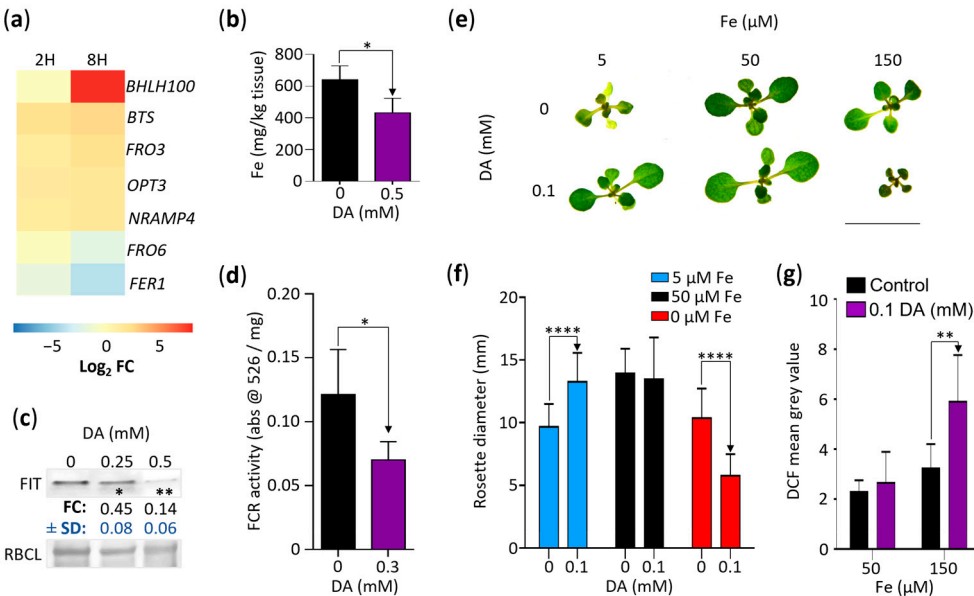

**Figure 8.** DA impacts iron homeostasis and generates elevated oxidative stress in the presence of high iron. (**a**) DA alters the expression of several iron-homeostasis-associated genes. Gene-level expression data were generated as described in the Materials and Methods. Iron-homeostasis-associated genes were extracted and are presented as a heatmap of log2 fold changes (Log2 FC) at 2 (2H) and 8 (8H) hours. All presented DEGs with a non-zero log2 FC have an FDR < 0.0001. (**b**) Plants grown on DA-supplemented media accumulate less iron. Col-0 seedlings grown for 7 days on control media and 0.5 mM DA were analyzed using ICP-MS (*n* = 3 pools of seedlings; * represents *p* < 0.05, Student's *t*-test). (**c**) DA represses the accumulation of FIT. Col-0 seedlings grown on MS/2 media supplemented with the denoted concentrations of DA for 7 days were used for immunoblotting analyses with anti-FIT antibodies. Representative immunoblot and the accompanying Ponceau S-stained membrane are shown (RBCL; large subunit of RuBisCO). Signal density was measured using ImageJ software, and fold change (FC) compared to the control ± SD is presented below the blots. (*n* = 3; * represents *p* < 0.05; ** represents *p* < 0.01; paired one-way ANOVA on raw density data with Dunnett's post-test). (**d**) DA-induced inhibition of ferric chelate reductase activity. Col-0 plants were grown for 12 days on MS/2 media supplemented with 0 or 0.3 mM DA. A total of ~10 mg of roots was excised and submerged in ferric chelate reductase (FCR) assay solution. After 25 min, the absorbance was measured at 526 nm. Data are presented as mean ± SD (*n* ≥ 5; * represents *p* < 0.05; one-way ANOVA with Tukey's post-test; comparisons are marked with one-way arrows). (**e**) Iron availability modulates the DA-induced inhibition of rosette growth. Col-0 seedlings grown for 14 days on iron-depleted media containing the denoted concentrations of DA and Fe-EDTA (Fe) were transferred to fresh plates and were photographed. Scale: 10 mm. (**f**) Quantification of the rosette diameter of plants grown as described in (**e**). Plants were photographed and rosette diameter was measured using ImageJ. Data are presented as mean ± SD. (*n* ≥ 30; **** represents *p* < 0.0001; one-way ANOVA with Tukey's post-test; comparisons are marked by one-way arrows). (**g**) Quantitative analysis of the influence of iron content in media on DA-induced ROS accumulation. Col-0 seedlings grown for 7 days on MS/2 media supplemented with the denoted concentrations of DA and Fe-EDTA were submerged in 10 μM H2DCF-DA and incubated at room temperature in complete darkness for 15 minutes. DCF was visualized using fluorescence microscopy. Mean grey value of DCF fluorescence was measured in 1.5 mm of root tissue (basipetal from root tip). Data are presented as mean ± SD. (*n* ≥ 9; ** represents *p* < 0.01; one-way ANOVA with Tukey's post-test; comparisons are demarked by one-way arrows).

An alternative hypothesis is that the combined pro-oxidant effects of iron and DA rather than the iron levels per se are key to DA-induced growth inhibition. It was reported previously that the overexpression of the γ-ECS encoding gene *GSH1* reduces the negative effects of iron deficiency [67], and we have presented evidence indicating that the DA

treatment of Arabidopsis seedlings increases GSH production (Figure 7a,b). Therefore, we concluded that the DA-dependent rescue of plants grown on low-iron media is likely a result of increased GSH biosynthesis, which is in agreement with the essential role of GSH in iron-deficiency tolerance [67].

Lastly, to determine if the accelerated oxidation of DA in the presence of iron [18] increases the ROS levels in Arabidopsis, we again employed H2DCF-DA. Indeed, the roots of the Col-0 seedlings grown on the media containing 150 μM Fe-EDTA and 0.1 mM DA for 7 days had higher DCF fluorescence than the roots of the high-iron control (Figure 8g). Together, our data suggest that the combined pro-oxidant activity of iron and DA is a primary cause of growth retardation, whereas changes in the iron homeostasis mechanisms are potentially a secondary response aimed at reducing iron availability and concomitantly minimizing ROS- and quinone-induced damage.

## 3. Discussion

Although DA has been detected in the extracts of many plant species, research on the DA metabolic pathway (e.g., the identities of the genes encoding biosynthetic enzymes or the identities of the enzymes responsible for the oxidation of DA and its conjugation to other phenolic compounds) is still ongoing [68]. In silico analyses showed that the Arabidopsis genome encodes for nine proteins with the DA-binding DA-ß-hydroxylase (DoH) domain [69], but whether these proteins function as DA receptors has not been tested. In addition, no specific DA transporters have been identified in plants. Therefore, we cannot unequivocally claim that DA functions as a signaling molecule in plants. Despite that, DA affects a myriad of processes in plants, including carbon metabolism, hormonal balance, and biotic and abiotic stress defense, to name a few [69–71]. Since these DA effects were investigated in different plant species, at different developmental stages, in different organs, and by using a range of DA concentrations and treatment methods, the lowest common denominator that can explain all the observed effects is the pro-oxidant property of DA itself.

Indeed, all the DA effects we investigated are linked to oxidative stress responses. For example, our GOA revealed that the genes associated with the response to oxidative stress were overrepresented among the upregulated DEGs (Table S2). The fact that, despite the increased antioxidative defenses, DA still leads to the inhibition of growth suggests that the DA-generated ROS load exceeds the antioxidative capacity of the cells. This conclusion is corroborated by our observation that treatment with the antioxidant GSH partially suppresses the decreased growth of Arabidopsis plants grown on DA-containing media (Figure 6c,d).

We also described two derived aspects of oxidative-stress-related growth inhibition. The first was enhanced auxin activity in the DA-treated plants caused partially by the DA-dependent upregulation of GSH biosynthesis (Figures 3, 6, 7 and S2). While we currently do not know how auxin activity is enhanced, our results show that auxin transport regulation has an important role (Figures 4, 5 and S3), which confirms a previous study showing that the positive effect of GSH on auxin activity involves the regulation of auxin transport [48]. The second derived effect of DA-induced oxidative stress was the dysregulation of iron homeostasis. Our transcriptomics analysis, iron content determination, and FCR activity measurements (Figure 8) suggest that the plants responded to the DA treatment by decreasing iron uptake. Separately, iron and DA are potent pro-oxidants, and, in combination, they lead to an accelerated generation of ROS and quinones, both of which can cause irreparable cell damage and cell death [18]. Indeed, a higher-than-optimal iron content caused plants to be more sensitive to the growth-inhibitory effects of DA (Figure 8). However, we could not conclude that the lower iron content of the DA-treated plants contributed to DA-induced growth inhibition. On the contrary, while the plants grown on the low-iron media were growth-inhibited, the addition of DA led to growth rescue (Figure 8e,f). This result corresponded well with previous studies showing that increased GSH counteracts the growth-inhibitory effects of iron deficiency [67].

Overall, our data show that DA induces a complex set of responses aimed at adapting to DA-induced oxidative stress and suggest that overcoming cellular oxidative stress defenses is the main mechanism of DA-dependent growth inhibition.

## 4. Materials and Methods

### 4.1. Growth Conditions and Materials

Plant lines used were Arabidopsis wild-type Col-0, *DR5::GUS* [72], *aux1-7* [73], *axr2-1* [74], *axr3-3* [41], and *axr5-2* [75]. *DR5::RFP* was generated by introducing the *DR5rev::erRFP* (AddGene, #61011) transgene into *TCSn::GFP* [76] (ABRC, CS69180) plants via the Agrobacterium floral dip method [77]. Plants were grown under sterile conditions in a controlled environmental chamber in continuous light (80 µmol m$^{-2}$ s$^{-1}$) at 24 °C on half-strength Murashige and Skoog media (Sigma–Aldrich, St. Louis, MO, USA) containing 1% sucrose and 0.8% phytoagar (MS/2, pH 5.7) or iron-depleted MS/2 media (Caisson, Smithfield, UT, USA) supplemented with Fe-EDTA (Sigma–Aldrich). The test compounds used in this study were reduced L-glutathione (GSH, Sigma–Aldrich), dopamine hydrochloride (DA, Acros Organics, New Jersey, USA), indole-3-acetic acid (IAA, PlantMedia, Dublin, OH, USA), 2-naphthaleneacetic acid (NAA, Sigma–Aldrich), and buthionine sulfoxamine (BSO, Enzo Life Sciences, Farmingdale, NY, USA).

### 4.2. GUS and Ferric Chelate Reductase (FCR) Assays

*DR5::GUS* seedlings were sown and grown for seven days on control and test media and were transferred to a buffer (10 mM Na$_2$EDTA, 10 mM NaH$_2$PO$_4$, 0.1% Triton X-100) containing X-Gluc substrate (1 mg/mL; GoldBio, St. Louis, MO, USA). Different incubation times were used for the GUS activity assays depending on the aim of the experiment. After staining, seedlings were cleared in ethanol and rehydrated in 50% (*v/v*) glycerol. For all experiments, a minimum of three biological replicates were carried out with a minimum of three seedlings per experiment. Seedlings representative of the experimental results are displayed.

The activity of root-associated FCRs was measured as described [78]. In brief, roots (~10 mg) of 12-day-old plants were excised and immersed into 1 mL of the assay solution (0.1 mM Fe-EDTA, 0.3 mM FerroZine; both from Sigma–Aldrich) and were incubated in the dark for 25 min at 24 °C. The absorbance of the assay solution was then measured at 526 nm and was normalized to root weight.

### 4.3. Fluorescent Microscopy

All micrographs were captured on an Olympus SZX12 microscope. For microscopic analyses of *DR5::RFP* expression, RFP fluorescence was captured using an excitation/emission filter of 545–580 nm/600–650 nm. For 2′,7′-dichlorodihydrofluorescein diacetate (H2DCF-DA; Biotium, Fremont, CA, USA ) experiments, seedlings were submerged in 10 µM H2DCF-DA and were incubated in the dark for 15 min. Seedlings were washed in distilled water, and 2′,7′-dichlorofluorescein (DCF) fluorescence was captured using an excitation/emission filter of 457–487/502–538 nm. For quantification of DCF and RFP fluorescence, bright-field and fluorescent images were stacked, roots were traced either 1.5 mm (for DCF signal) or 600 µm (for RFP signal) from the root tip, and mean gray values were measured using ImageJ software.

### 4.4. RNAseq Analysis

Col-0 was grown vertically for 7 days on MS/2 media and was incubated in water or 50 mM DA for 2 or 8 h. The tissue was washed, blotted dry, weighed, and snap frozen in liquid nitrogen. RNA extraction, library construction, and sequencing were carried out by Novogene (Sacramento, CA, USA). In brief, mRNA was extracted and purified using poly-T oligo-attached magnetic beads and was fragmented. First-strand cDNA was synthesized using random hexamer primers followed by second-strand cDNA synthesis. Library then underwent end repair, A-tailing, adapter ligation, size selection, amplification,

purification, and sequencing using the Illumina Hiseq 2000 platform. Errors containing k-mers were corrected using rCorrecter [79], and unfixable read pairs were removed. Ribosomal RNA contamination was removed with mapping reads using bowtie2 to a ribosomal RNA blacklist generated from SILVA [80]. Data was then analyzed by comparing the DA treatment to the water control at each timepoint using Salmon [81] in GC bias-aware mode using the TAIR10 transcriptome as an index. Gene-level expression data were generated from Salmon output using tximport [82], and differential expression analysis was performed using EdgeR [83]. DEGs with a Log2 fold change > 1 and <−1 and an FDR < 0.05 were considered significant. Heatmaps were produced using the R package pheatmap. GOA was performed using PANTHER [84] (database: 10.5281/zenodo.6799722) using a list of all genes subjected to differential expression analysis at each individual timepoint as a background (see Reference Lists in Table S2). Fisher's exact test was used with a cutoff of FDR < 0.05.

Auxin metabolic pathway and associated gene sets were curated from a combination of literature and the Plant Metabolic Network [34,35]. Process-specific gene sets used to support experimental data were generated from Gene Ontology (GO) groups extracted from ThaleMine [85]. Heatmaps with their corresponding GO groups are as follows: Figure S2a—response to auxin, 0009733; Figure S3—auxin influx transmembrane transporter activity, 0010328; auxin efflux transmembrane transporter activity, 0010329; Figure 6a—glutathione biosynthetic process, 0006750; glutathione metabolic process, 0006749; Figure 8a—cellular iron ion homeostasis, 0006879; and ferric chelate reductase activity, 0000293.

### 4.5. Immunoblotting Analysis

The 7-day-old plants were snap frozen in liquid nitrogen and were disrupted using zirconium beads in 2 volumes of extraction buffer (200 mM sucrose, 2.4 mM sodium deoxycholate, 50 mM Tris-HCL with pH of 7.5, 1 mM EDTA, 0.1 mM phenylmethylsulfonylfluoride). Protein concentration was measured using Quick Start Bradford protein assay kit (Bio-Rad) and using bovine serum albumin as a standard and was equalized between samples using the extraction buffer. Immunoblotting analyses were done as described [86] using anti-FIT1 (PhytoAB, San Jose, CA, USA; 1:2000) and goat anti-rabbit IgG-HRP (Santa Cruz Biotech, Dallas, TX, USA; 1:1000). Immunoblots were developed using SuperSignal West Femto Maximum Sensitivity Substrate (Thermo Fisher Scientific, Waltham, MA, USA). Signals were captured using ChemiDoc XRS (Bio-Rad, Hercules, CA, USA) and were then quantified using ImageJ software, version 1.53a.

### 4.6. Auxin Metabolite Profiling

Col-0 seedlings were grown for 7 days on control or 0.5 mM DA media. Three replicates of 1 g each were harvested and snap frozen in liquid nitrogen. UPLC-ESI-MS/MS metabolomic profiling was done by Lifeasible (Shirly, NY 11967). Samples were ground into powder, were extracted in 15:4:1 methanol/water/formic acid, and were evaporated to dryness under nitrogen gas steam. Pellets were reconstituted in 80% methanol, were filtered (0.22 μm), and were analyzed via UPLC-ESI-MS/MS (UPLC, ExionLC AD; MS, Applied Biosystems 6500 Triple Quadrupole) equipped with a Waters ACQUITY UPLC HSS T3 C18 column. Solvent systems were 0.04% acetic acid (A) and acetonitrile containing 0.04% acetic acid (B). Gradient started at 5% B (0–1 min), ramped up to 95% B (1–8 min), stayed at 95% B (8–9 min), and finally ramped down back to 5% B (9.1–12 min). Flow rate was 0.35 mL/min, and temperature was 40 °C. Mass spectrometry was done using an AB 6500+ QTRAP® LC-MS/MS System equipped with an ESI Turbo Ion–Spray interface, operating in both positive and negative ion modes, and controlled by Analyst 1.6 software (AB Sciex). Operation parameters were as follows: ion source, turbo spray, source temperature of 550 °C, ion spray voltage of (IS) 5500 V (positive) and −4500 V (negative), curtain gas set at 35.0 psi, and declustering potential (DP) and collision energy (CE) for individual multiple reaction monitoring (MRM) transitions done with further DP and CE optimization.

A specific set of MRM transitions were monitored for each period according to the plant hormones eluted within this period. Analyst 1.6.3 was used to process mass spectrum data based on a local metabolic database, and quantitative analysis was carried out using the mass spectrum peak intensity data of the corresponding signal of the standard curves of 17 different auxin metabolites. Metabolites with a fold change greater than 2 and less than 0.5 compared to the control were considered differentially accumulating.

### 4.7. Iron Determination

Col-0 seedlings were grown for 7 days on media with no DA or 0.5 mM DA. Three replicates of each treatment (5 g each) were washed in 0.01% Triton X-100, were rinsed with distilled water, were snap frozen, and were analyzed by Lifeasible (Shirly, NY 11967) using ICP-MS. In brief, samples were pressure-tank-digested using nitric acid and were heated until nearly dry. Digestive liquid was washed in a volumetric flask using nitric acid, and iron content was determined using a NexION2000 ICP-MS (PerkinElmer, Waltham, MA, USA).

### 4.8. Data Analysis

Statistical analyses of growth responses and fluorescent and immunoblotting signal intensities were conducted using GraphPad Prism 6. All data were obtained from three or more biologically independent replicates. Statistical comparisons were done as described in the main text and legends.

**Supplementary Materials:** The following supporting information can be downloaded at https://www.mdpi.com/article/10.3390/stresses3010026/s1, Figure S1: Plants grown on DA have increased lateral root length and adventitious root number, Figure S2: DA increases auxin activity, Figure S3: Differential expression of genes associated with auxin transport in response to DA exposure, Table S1: DEGs in response to 50 mM dopamine at 2 and 8 h, Table S2: Results of Gene Ontology enrichment analysis, and Table S3: Results of targeted UPLC-ESI-MS/MS analysis of plants grown on 0 or 0.5 mM DA.

**Author Contributions:** Conceptualization, T.E.S., J.A.S. and J.K.; methodology, T.E.S., J.A.S. and J.K.; software, T.E.S.; validation, T.E.S., J.A.S. and J.K.; formal analysis, T.E.S.; investigation, T.E.S.; resources, T.E.S., J.A.S. and J.K.; data curation, T.E.S.; writing—original draft preparation, T.E.S.; writing—review and editing, J.A.S. and J.K.; visualization, T.E.S.; supervision, J.A.S. and J.K.; project administration, T.E.S. and J.A.S.; funding acquisition, T.E.S. and J.A.S. All authors have read and agreed to the published version of the manuscript.

**Funding:** This work was funded by the USDA/NIFA pre-doctoral fellowship #2020-67034-31753 of T. E. Shull and the Kentucky Tobacco Research and Development Center. The graphical abstract and models were produced using BioRender under an academic license owned by T. E. Shull at the time of publication.

**Data Availability Statement:** The raw RNA sequencing reads discussed in this manuscript were deposited in the NCBI sequence read archive under BioProject PRJNA903201. All other data presented in this study are available in the figures and supplemental files provided in the manuscript.

**Conflicts of Interest:** The authors declare no conflict of interest. The funders had no role in the design of the study; in the collection, analyses, or interpretation of the data; in the writing of the manuscript; or in the decision to publish the results.

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
