# Peer review of "Dopamine Inhibits Arabidopsis Growth through Increased Oxidative Stress and Auxin Activity"

_stresses, doi:10.3390/stresses3010026_

Round 1
Reviewer 1 Report
The authors in this manuscript describe some biological roles of dopamine in plants and in particular Arabidopsis thaliana. They use pharmacological, physiological and molecular approaches to show that exogenous application of dopamine increases sensitivity to IAA by increasing auxin responses, induces oxidative stress and downregulates iron uptake mechanisms. The manuscript is overall well written and presented.
Some issues that should be addressed though:
-line 24 is [1] the most appropriate reference for that?
-the introduction could be enriched a little bit more about the role of DA in plants
-line 93 ''To assess the most impacted processes associated with DA-regulated DEGs....'' in the RNA-seq analysis the authors state that they found 1152 differentially expressed genes, but they focus on only the top 4 upregulated and downregulated. The authors should provide at least a GO enrichment analysis of their findings to assess if other processes are also affected, the GO enrichment analysis should also show that auxin, oxidative stress and iron uptake to be affected by DA.
-line 130, the authors should present the data with the DOA gene expressions
-in 2.2 part the manuscript would benefit if the authors showed the expression of other auxin biosynthesis genes such as TAA1 and the YUCCA family members
-line 161 are there more auxin downregulated genes that are also downregulated by DA? one example may not be enough
-2.4 part, the authors could test possible synergistic effects of DA with NPA, a polar auxin transport inhibitor, in root elongation assays, to further support their claim that the DA/IAA interaction requires auxin transport machinery
Reviewer 2 Report
please find the attached review report.

Round 2
Reviewer 1 Report
Dear authors
I am pleasantly surprised from the well written and to-the-point cover letter you provided about my comments. You change the things you believe will benefit the manuscript and provide good points for not changing other things, including points i have missed my self. I am satisfied about the changes you have done to the manuscript and i believe the result is better than the original draft.